# A Case of Milk-Alkali Syndrome Caused by Diuretic-Induced Alkalosis and Polypharmacy

**DOI:** 10.3390/medicina59071345

**Published:** 2023-07-22

**Authors:** Naoya Mizutani, Ken Goda, Tsuneaki Kenzaka

**Affiliations:** 1Department of Internal Medicine, Hyogo Prefectural Tamba Medical Center, Tamba 669-3495, Japan; naomizutaning@gmail.com (N.M.); kenkenpetneed@yahoo.co.jp (K.G.); 2Division of Community Medicine and Career Development, Kobe University Graduate School of Medicine, Chuo-ku, Kobe 650-0017, Japan

**Keywords:** milk-alkali syndrome, hypercalcemia, metabolic alkalosis, renal dysfunction, comorbidities, loop diuretics, case report

## Abstract

Milk-alkali syndrome, which is characterized by hypercalcemia, metabolic alkalosis, and renal dysfunction, typically results from the ingestion of large amounts of calcium and absorbable alkaline products. However, these symptoms can also manifest when alkalosis and calcium loading occur simultaneously, owing to other factors. We report a case of milk-alkali syndrome caused by loop-diuretic-induced alkaline load and polypharmacy in an 85-year-old Japanese woman with multiple comorbidities, including osteoporosis, hypertension, type 2 diabetes, dyslipidemia, and Parkinson’s disease. The patient regularly took 14 drugs, including calcium L-aspartate, eldecalcitol, celecoxib, and a fixed-dose combination of losartan and hydrochlorothiazide. Immediately before admission, furosemide was administered for the treatment of edema. The patient presented with chest discomfort, general malaise, and clinical signs of dehydration, hypercalcemia, hypophosphatemia, hypokalemia, and hypomagnesemia, accompanied by electrocardiogram abnormalities, renal dysfunction, and chloride-resistant metabolic alkalosis. The hypercalcemia was specifically induced by calcium L-aspartate and eldecalcitol. The hypomagnesaemia and hypophosphatemia were caused by diuretics and hypercalcemia. Thus, all the oral medications were discontinued, and rehydration and electrolyte correction therapy were administered. The final diagnosis was milk-alkali syndrome caused by the concomitant use of loop diuretics and other medications, without absorbable alkaline preparation use. This case underscores the importance of considering drug-related factors, checking concomitant medications, and being aware of the benefits, harmful effects, and side effects of polypharmacy in older adults with multimorbidity.

## 1. Introduction

Milk-alkali syndrome is characterized by hypercalcemia, metabolic alkalosis, and renal dysfunction. This condition is associated with the ingestion of large amounts of calcium and absorbable alkaline preparations [1]. Milk-alkali syndrome is typically reported as an adverse event resulting from the ingestion of milk and sodium bicarbonate (absorbable alkaline preparation), which is commonly used to treat gastric ulcers. In recent years, a similar pathology has been reported to occur due to a vicious metabolic cycle involving calcium preparations and alkaline loading, such as magnesium oxide [2]. Hypercalcemia, metabolic alkalosis, and renal dysfunction can occur if alkalosis and calcium loading occur simultaneously, owing to other factors [3].

Herein, we report a case of milk-alkali syndrome in an older female patient who had several comorbidities, and was regularly taking numerous medications. The patient syndrome occurred due to an alkaline load induced by loop diuretics, even without the use of absorbable alkaline preparations.

## 2. Case Presentation

An 85-year-old Japanese woman was brought to our emergency department, having had complaints of chest discomfort and general malaise for three days. Her comorbidities included osteoporosis, hypertension, type 2 diabetes, dyslipidemia, and Parkinson’s disease. She had visited the departments of internal medicine, psychiatry, and orthopedics at hospital A and clinic B for the management of these diseases. She was able to independently perform her daily life activities. She had no cognitive impairment, and could self-manage her medications, which were as follows: calcium L-aspartate (400 mg/day), eldecalcitol (0.75 μg/day), celecoxib (200 mg/day), rebamipide (200 mg/day), levodopa-benserazide (2 tablets/day), famotidine (40 mg/day), pirenzepine (50 mg/day), vaccinia virus-inoculated rabbit inflammatory skin extract (16 units/day), sitagliptin (50 mg/day), ethyl icosapentate (900 mg/day), losartan (50 mg)/hydrochlorothiazide (12.5 mg) combination (1 tablet/day), atorvastatin (10 mg/day), ezetimibe (10 mg/day), and zolpidem (5 mg). The zolpidem was prescribed to be taken using the ‘pill-in-the-pocket’ approach. Thus, the patient was taking 14 oral medications, all of which she took regularly, in appropriate doses. Two weeks before admission, her physician in the psychiatry department of hospital A added furosemide (40 mg/day) to her medications, to treat leg edema. An examination performed three days prior to admission revealed anemia, and a right ureteral stone. Multiple gastric ulcers (A2 stage) were noted on an upper gastrointestinal endoscopy the day before admission.

On examination, the patient showed clear consciousness, and her vital signs were as follows: blood pressure, 114/61 mmHg; pulse rate, 64 beats per minute and regular; body temperature, 36.3 °C; respiratory rate, 19 breaths per minute; and oxygen saturation, 95% while breathing ambient air. Her general appearance indicated illness. In addition, she had a curved back. She had no eyelid or conjunctival pallor. Her mouth and axillae were dry, and her capillary refilling time was at least 5 s. No significant abnormal breath sounds, heart sounds, or abdominal findings were observed. Her superficial lymph nodes were not palpable, and she had no leg edema. Her main blood test results are presented in Table 1. The results of her urinalysis were as follows: K, 22.2% (10–20); P, 30.0% (10–20); Mg, 4.2% (2–3); urinary K/Cr ratio, 25.4 mEq/g·Cr; and urinary Ca excretion, 0.32 mg/dLGF (Table 1). A 12-lead electrocardiogram (ECG) revealed the following findings: heart rate, 59 beats/min; rhythm, sinus rhythm; T wave, flat-low; and QTc, 0.53 s. Computed tomography (CT) without chest enhancement showed no enlarged lymph nodes in the mediastinum or hilar region. Additionally, there were no signs of pleural or pericardial effusion. An abdominal CT scan (Figure 1) without enhancement showed multiple bilateral urinary tract stones. The abdominal CT also revealed hydronephrosis and hydroureteral disease on the left side. There were no enlarged lymph nodes in the para-aortic region or pelvis. Significant hypercalcemia, hypophosphatemia, hypokalemia, and hypomagnesemia, accompanied by ECG abnormalities, renal dysfunction, and chloride-resistant metabolic alkalosis were observed.

The results of the tests related to hypercalcemia (Table 1) did not indicate hyperparathyroidism or sarcoidosis. In addition, there was no evidence of malignancy or bone metastases on the CT or endoscopy. The venous blood gas analysis results revealed remarkable metabolic alkalemia. The patient’s hypochloremia was indicative of chloride depletion and contraction alkalosis. Further, the patient’s urine potassium/creatinine ratio was over 13 mEq/g·Cr, her blood pressure was not high, and her urine Cl was over 15 mEq/L, which was suggestive of active diuretic use, or a hereditary tubular disease, such as Bartter syndrome or Gitelman syndrome [4]. Based on the history of the present illness, we suspected that the patient’s symptoms were likely caused by diuretic-induced hypokalemia with hypomagnesemia, which maintains a Cl-resistant metabolic alkalemia. Therefore, we discontinued all her oral medications, and performed electrolyte correction by infusion. The electrolyte and acid–base imbalance improved by day 6 of admission (Figure 2). The subsequent discontinuation of the electrolyte correction did not worsen the electrolyte imbalance, acid–base imbalance, or renal dysfunction. Other than drug-induced hypercalcemia attributable to the use of calcium L-aspartate and eldecalcitol, no significant findings regarding the cause of the hypercalcemia were identified. The patient’s chronic hypercalcemia was considered to have caused the renal dysfunction, and multiple gastric ulcers. Concomitant alkaluria and hypercalciuria were considered the causes of the ureteral stones. In addition, the anemia was considered to be caused by bleeding from the multiple gastric ulcers. The hypomagnesemia and hypophosphatemia were attributed to combination therapy using two types of diuretics, specifically furosemide with hydrochlorothiazide, and the effect of the hypercalcemia. The hypokalemia was also considered to be influenced by the diuretics. No hypertension was observed during hospitalization, and no antihypertensive drugs were administered.

The patient was finally diagnosed with milk-alkali syndrome due to the concomitant use of loop diuretics and regular use of thiazide diuretics, active vitamin D preparations, calcium preparations, angiotensin II receptor blockers, and COX-2 selective inhibitors. She was discharged 15 days after admission, and her prescription at discharge was reduced to four drugs: levodopa-benserazide, potassium chloride, magnesium oxide, and rabeprazole. Calcium L-aspartate and eldecalcitol were not continued, due to concerns regarding recurrent hypercalcemia. The patient has been under outpatient observation for 16 months since her discharge. The potassium chloride and magnesium oxide were reduced or discontinued, and amlodipine was introduced into her regimen. The patient’s electrolyte and acid–base equilibria remain within normal ranges, without correction.

## 3. Discussion

We report a case of milk-alkali syndrome caused by polypharmacy in an older female patient with comorbidities, including osteoporosis and hypertension. The patient was taking numerous concomitant medications for these comorbidities, including loop diuretics and thiazides. The diuretics induced an alkaline load that caused milk-alkali syndrome, even without the use of absorbent alkaline preparations. There have been previous reports of milk-alkali syndrome caused by alfacalcidol and thiazide [1]. However, to the best of our knowledge, this is the first report of milk-alkali syndrome caused by polypharmacy including both loop diuretics and thiazides.

Aging causes changes to many organs, including the kidney, which maintains the electrolyte balance. Diuretic use may impair the ability to maintain homeostasis, resulting in susceptibility to electrolyte and acid–base disorders [5].

Diuretics, which deliver an alkaline load, can be a precipitating factor in the development of milk-alkali syndrome. Loop diuretics promote urinary calcium excretion, and are used to treat hypercalcemia. In the present case, the loop diuretic induced an alkaline load, causing milk-alkali syndrome. The patient was subjected to polypharmacy, due to several co-existing comorbidities. Notably, she was taking celecoxib (a COX-2 selective inhibitor) and losartan (an angiotensin II receptor blocker (ARB)), both of which reduce the renal blood flow. In addition, the patient was also on hydrochlorothiazide, which reduces the urinary calcium excretion, and contributes to hypercalcemia [6,7]. Hydrochlorothiazide can also predispose a patient to hypomagnesaemia and hypokalemia. Thiazide diuretics increase proton secretion from the cortical collecting ducts, and cause metabolic alkalosis, resulting in an alkaline load. A reduced glomerular filtration rate stimulates bicarbonate reabsorption in the proximal tubules, and maintains metabolic alkalosis [8,9,10,11,12]. Thiazides can also cause profound hypercalcemia without excess calcium intake, hyperparathyroidism, or elevated vitamin D levels [13]. Furthermore, calcium L-aspartate and eldecalcitol, which the patient was taking as treatment for osteoporosis, both increase the serum calcium concentration. The concomitant use of celecoxib, losartan, hydrochlorothiazide, calcium L-aspartate, and eldecalcitol maintained iatrogenic hypercalcemia, by increasing the renal calcium absorption through contraction alkalosis, and decreasing the glomerular filtration rate and renal calcium excretion. Furosemide, a loop diuretic, may have added to the alkaline load, enhancing the vicious metabolic cycle, and leading to the manifestation of the patient’s symptoms.

The initial treatment for hypercalcemia is saline infusion therapy [14]. There is no evidence that diuretics exacerbate alkalosis or renal dysfunction, as long as adequate infusions are administered [15]. Loop diuretics are contraindicated for the treatment of hypercalcemia in cases of volume depletion, as this worsens the hypercalcemic contraction alkalosis.

The frequent use of a combination of several medications for the treatment of multiple comorbidities may result in milk-alkali syndrome. Milk-alkali syndrome is suggested to be the third-most-common cause of hypercalcemia requiring hospitalization, with a high proportion of the cases being severe in intensity [16]. There are also previous reports of milk-alkali syndrome complicated by secondary acute pancreatitis [17,18] and acute encephalopathy [19]. Milk-alkali syndrome should be considered a differential diagnosis in patients with hypercalcemia and multiple comorbidities.

According to the American Geriatrics Society Beers Criteria^®^ (AGS Beers Criteria^®^) [20], potentially inappropriate medications (PIMs) are characterized by the following features: (1) medications that are potentially inappropriate for most older adults; (2) inappropriate medication use in older adults due to drug–disease or drug–syndrome interactions that may exacerbate the disease or syndrome; (3) drugs to be used with caution in older adults; (4) potentially clinically important drug–drug interactions that should be avoided in older adults; and (5) medications that should be avoided, or should have their dosages reduced, for older adults with varying levels of kidney function. The polypharmacy in the present case demonstrated features similar to those of PIMs. While the AGS Beers Criteria^®^ mention diuretics and COX-2 selective inhibitors, osteoporosis drugs are not listed. Medications for osteoporosis are classified as potential prescribing omissions (PPOs) in the STOPP/START criteria version 2 [21]. A previous report described the potentially inappropriate use of NSAIDs and PPO of osteoporosis drugs, based on the STOPP/START criteria version 1, in older patients residing at home in Japan [22]. However, in the present case, the patient was concomitantly on both these types of drugs. It should be noted that even when the explicit aforementioned criteria are followed, drug–drug interactions, specifically involving PIMs, may amplify the undesirable effects of some medications, and elevate the risk of adverse drug reactions [23], by upsetting the balance between the beneficial and harmful effects of prescription drugs.

Adverse drug events are important factors in the admission of patients to acute care hospitals [24]. Moreover, PIMs and severe drug interactions are relatively common in hospital care [23]. This may be because PIMs comprising prescriptions from several medical institutions are not recognized until emergency admission to acute care hospitals. In older patients with multimorbidity, PIMs should be comprehensively evaluated, and adverse drug events always considered. Despite evidence that polypharmacy is associated with an increased risk of undesirable outcomes, such as falls, cognitive impairment, hospitalizations, medication burden, and increased healthcare costs among older adults, deprescribing has not been widely embraced [25]. One systematic review suggested the comparative effectiveness of deprescribing interventions in community-dwelling individuals aged 65 or older. Thus, a comprehensive medication review in such patients may reduce mortality, and three types of deprescribing interventions, including provider and/or patient education, and computerized decision support, may reduce the number of PIMs prescribed [26].

## 4. Conclusions

In this report, we described the case of an older female patient with several comorbidities, including osteoporosis and hypertension, for which she was taking numerous medications. These medications included loop diuretics, COX-2 selective inhibitors, angiotensin II receptor blockers (ARBs), thiazide diuretics, calcium L-aspartate, and eldecalcitol, and their concomitant use resulted in an alkaline load and milk-alkali syndrome. In patients with multimorbidity, the frequent use of a combination of drugs and drug interactions can lead to the development of milk-alkali syndrome. When treating older adults, it is important to always consider drug-related factors, check concomitant medications, and be aware of the benefits and harmful side effects of polypharmacy.

## Figures and Tables

**Figure 1 medicina-59-01345-f001:**
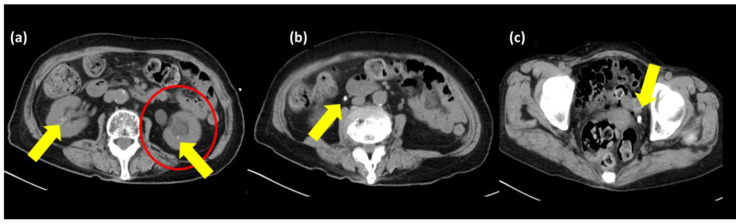
An abdominal computed tomography scan without enhancement. The images are in the order (**a**–**c**). Bilateral kidney and ureteral stones (yellow arrows). Left-sided hydronephrosis and left-sided hydroureter (red circle).

**Figure 2 medicina-59-01345-f002:**
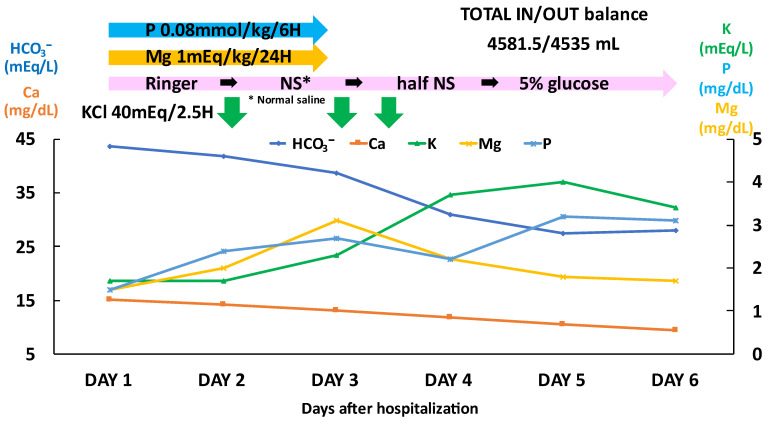
The course of treatment for the first six days after hospitalization.

**Table 1 medicina-59-01345-t001:** Laboratory data upon admission, and on day 2.

Parameter	Recorded Value	Standard Value/Reference Range
White blood cell count	9100/µL	4500–7500/µL
Hemoglobin	9.3 g/dL	11.3–15.2 g/dL
Platelet count	7.5 × 10^4^/µL	13–35 × 10^3^/µL
C-reactive protein	0.24 mg/L	≤0.60 mg/dL
Total protein	6.9 g/dL	6.9–8.4 g/dL
Albumin	4.2 g/dL	3.9–5.1 g/dL
Total bilirubin	0.7 mg/dL	0.2–1.2 mg/dL
Aspartate aminotransferase	30 U/L	11–30 U/L
Alanine aminotransferase	9 U/L	4–30 U/L
Lactase dehydrogenase	226 U/L	109–216 U/L
Creatine kinase	273 U/L	40–150 U/L
Blood urea nitrogen	42.3 mg/dL	8–20 mg/dL
Creatinine	1.49 mg/dL	0.63–1.03 mg/dL
Sodium	145 mEq/L	136–148 mEq/L
Potassium	1.7 mEq/L	3.6–5.0 mEq/L
Chloride	91 mEq/L	98–108 mEq/L
Calcium	15.2 mg/dL	8.8–10.1 mg/dL
Phosphorus	1.5 mg/dL	2.7–4.6 mg/dL
Magnesium	1.5 mg/dL	1.8–2.6 mg/dL
Glucose	144 mg/dL	70–109 mg/dL
Hemoglobin A1c	5.1%	5.6–5.9%
Tests related to hypercalcemia (day 2)		
Angiotensin-converting enzyme	6.8 U/L	7.0–25.0 U/L
1,25-dihydroxyvitamin D3	27 pg/mL	20.0–60.0 pg/mL
Intact parathyroid hormone	14 pg/mL	15.0–65.0 pg/mL
Parathyroid hormone-related protein-C	<1.1 pmol/L	<1.1 pmol/L
Venous blood gas analysis		
pH	7.546	
pCO_2_	50.3 mmHg	
HCO_3_^−^	43.6 mEq/L	
Base excess	19.0 mEq/L	
Urinalysis		
pH	7.0	
Specific gravity	1.011	
Protein	+/−	
Occult blood	2+	
Sodium	37 mEq/L	
Potassium	18.5 mEq/L	
Chloride	30 mEq/L	
Calcium	15.7 mg/dL	
Phosphorus	22.0 mg/dL	
Magnesium	3.1 mg/dL	

## Data Availability

All data generated or analyzed for this report are included in the published article.

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
