# Peer review of "A Case of Milk-Alkali Syndrome Caused by Diuretic-Induced Alkalosis and Polypharmacy"

_medicina, 2023, doi:10.3390/medicina59071345_

Round 1

Reviewer 1 Report

1.     I believe that the cause of milk-alkali syndrome in the presented case is due to the combination of calcium supplements, thiazide diuretics, NSAID, and angiotensin-converting enzyme inhibitors (ACEI). Thiazide diuretics can cause volume depletion and contraction alkalosis, increasing renal calcium absorption. ACEI and NSAIDs decrease renal calcium excretion by decreasing the GFR.

2.     Furosemide was only recently introduced into the patient's medication regimen in a moderate dose. Moreover, the presence of urinary lithiasis suggests a more extended process. Please reduce emphasis on furosemide as a cause of milk-alkali syndrome.

3.     Please report the vital signs and only the pathological clinical exam findings. A table with the blood tests would be helpful and easier to follow

4.     A differential diagnosis of hypercalcemia in this age group is recommended to be made and presented in the manuscript - perhaps in a table format with the tests used for diagnostic exclusion.

5.     How did you perform electrolyte correction with transfusions

6.     A graph demonstrating the dynamic values and the administered treatment would be very informative.

7.     The authors used a treatment strategy known as "deprescribing"; please discuss this in relation to polypharmacy.

8.     Milk-alkali syndrome is not the third most common cause of hypercalcemia requiring hospitalization, at least not globally; the provided reference is somewhat outdated and focused on the US.

9.     The authors do not discuss the causes and management of urinary lithiasis.

 Moderate editing of English language required

Author Response

  1. I believe that the cause of milk-alkali syndrome in the presented case is due to the combination of calcium supplements, thiazide diuretics, NSAID, and angiotensin-converting enzyme inhibitors (ACEI). Thiazide diuretics can cause volume depletion and contraction alkalosis, increasing renal calcium absorption. ACEI and NSAIDs decrease renal calcium excretion by decreasing the GFR.

    Response:
    Thank you for your comment. As you have mentioned, herein, volume depletion caused by thiazide diuretics resulted in contraction alkalosis, which increased renal calcium absorption. In addition, the COX-2 inhibitor and angiotensin II receptor blocker prescribed to the patient decreased the glomerular filtration rate and renal calcium excretion. This, in addition to the patient’s calcium supplement use, maintained iatrogenic hypercalcemia. We have added this information in the manuscript as follows (lines 159–162):

    Original:
    “The combination of these five drugs caused hypercalcemia and volume depletion due to metabolic alkalosis.”

    Revised:
    “Concomitant use of celecoxib, losartan, hydrochlorothiazide, calcium L-aspartate, and eldecalcitol maintained iatrogenic hypercalcemia by increasing renal calcium absorption through contraction alkalosis and decreasing the glomerular filtration rate and renal calcium excretion.”
  2. Furosemide was only recently introduced into the patient's medication regimen in a moderate dose. Moreover, the presence of urinary lithiasis suggests a more extended process. Please reduce emphasis on furosemide as a cause of milk-alkali syndrome.

    Response:
    Thank you for your comment. We should have mentioned that iatrogenic hypercalcemia in this case was chronically maintained by five drugs: celecoxib (a COX-2 selective inhibitor), losartan (an angiotensin II receptor blocker), hydrochlorothiazide, calcium L-aspartate, and eldecalcitol, before furosemide was added to the patient’s regimen. This information has been added in lines 159–162, as mentioned above.
    We believe that furosemide worked as a recent and direct cause of the manifestation of symptoms in this case. Thus, we had emphasized furosemide as a cause of the milk alkali syndrome in the present case.
    In order to reduce the emphasis on furosemide, we have revised the text as follows:

    Original:
    “The patient was taking numerous concomitant medications for these comorbidities, including loop diuretics. The loop diuretics induced an alkaline load that resulted in the milk-alkali syndrome, even without the use of absorbent alkaline preparations.”

    Revised (underlined; line 135):
    “The patient was taking numerous concomitant medications for these comorbidities, including loop diuretics and thiazides. The diuretics induced an alkaline load that resulted in the milk-alkali syndrome, even without the use of absorbent alkaline preparations.”

    We have also added the following sentence to clarify the relationship between hypercalcemia and diuretics-inducible contraction alkalosis:

Revised (underlined; lines 168–169):
“Loop diuretics are contraindicated for the treatment of hypercalcemia in cases of volume depletion, as this worsens hypercalcemic contraction alkalosis.”

  1. Please report the vital signs and only the pathological clinical exam findings. A table with the blood tests would be helpful and easier to follow.

    Response:
    Thank you for your advice. The patient’s vital signs are reported in lines 67–69. We have also added Table 1 to present the laboratory data upon admission and on day 2. The corresponding information was removed from the text to avoid duplication of information.
  2. A differential diagnosis of hypercalcemia in this age group is recommended to be made and presented in the manuscript - perhaps in a table format with the tests used for diagnostic exclusion.

    Response:
    Thank you for your comment. In lines 92–94, we have mentioned hyperparathyroidism, sarcoidosis, and malignancy or bone metastases as differential diagnoses for hypercalcemia in older adults that were ruled out.
  3. How did you perform electrolyte correction with transfusions?

    Response:
    Thank you for your comment. We have added Figure 2 in response to your comments. The figure depicts the course of treatment in this case with the dynamic values and is cited in line 104.
    We have also replaced the term “transfusions” with “infusion” on line 103, based on the recommendation by Reviewer 2.

  4. A graph demonstrating the dynamic values and the administered treatment would be very informative.

    Response:
    Thank you for your suggestion. As mentioned in the previous response, we have added Figure 2, which is a graph depicting the course of treatment in this case, including the dynamic values. The figure is cited in line 104.

  5. The authors used a treatment strategy known as "deprescribing"; please discuss this in relation to polypharmacy.

    Response:
    Thank you for your comment. We have added the relevant information in the Discussion section and included two supporting references (numbers 25 and 26). The revised text is as follows (lines 201–209):

“Despite evidence that polypharmacy is associated with an increased risk of undesirable outcomes, such as falls, cognitive impairment, hospitalizations, medication burden, and increased healthcare costs among older adults, deprescribing has not been widely embraced [24]. One systematic review suggested the comparative effectiveness of deprescribing interventions in community-dwelling individuals aged 65 or older. Thus, a comprehensive medication review in such patients may reduce mortality, and three types of deprescribing interventions, including provider and/or patient education and computerized decision support, may reduce the number of PIMs prescribed [26].”

  1. Milk-alkali syndrome is not the third most common cause of hypercalcemia requiring hospitalization, at least not globally; the provided reference is somewhat outdated and focused on the US.

    Response:
    Thank you for your indication. We have retained reference number 15 to emphasize that iatrogenic hypercalcemia resulting in severe milk-alkali syndrome is more common than expected.
    We have made the following change to avoid conclusive expression:

Original
“Milk-alkali syndrome is the third most common cause of hypercalcemia requiring hospitalization, with a high proportion of the cases being severe in intensity [15].”

Revised (underlined; lines 171–172):
“Milk-alkali syndrome is suggested to be the third most common cause of hypercalcemia requiring hospitalization, with a high proportion of the cases being severe in intensity [16].”

  1. The authors do not discuss the causes and management of urinary lithiasis.

    Response:
    Thank you for your valuable comment. We have added the recorded value of urinary pH in Table 1 and have revised the corresponding text as follows:

    Original
    “The renal dysfunction, multiple gastric ulcers, and ureteral stones were considered to be affected by hypercalcemia.”

    Revised (lines 107–110):
    “The patient’s chronic hypercalcemia was considered to have caused the renal dysfunction and multiple gastric ulcers. Concomitant alkaluria and hypercalciuria were considered causes of the ureteral stones.”
  2. Moderate editing of English language required

The revised manuscript has been proofread by a native English speaker.

Reviewer 2 Report

 A case of milk-alkali syndrome due to loop diuretic-induced alkaline load and polypharmacy by Naoya Mizutani and colleagues is a good and instructive review paper that exemplifies well the potential consequences of polypharmacy especially in the elderly.
However, individual points should be reiterated:
1. the determination of the chloride concentration in serum and urine is indispensable for the differential diagnosis of metabolic alkalosis (and basically also for urine ammonium excretion).
2. The amount and type of infusion therapy (ideally in milliliters per kilogram of body weight and the time period) shoulbe added. A follow-up regarding the urine could drip would be desirable.
3. line 107: instead of "transfusion" --> infusion should be meant.
4. in the discussion it should be mentioned that especially elderly people with acute renal deterioration are considerably vulnerable to higher electrolyte shifts (Khow, Lau et al. 2014).
5. line 137/138: I think this sentence should be deleted. Loop diuretic therapy has been shown to force contraction alkalosis, but it has not been adequately or sufficiently proven as the sole cause (even in this case). Moreover, there are other publications that shed more light on this point of view (Bohmig, Schmaldienst et al. 1999).
6. the title should replace the term "alkaline load" by solely alkalosis, since pathophysiologically primarily the contraction alkalosis by the intensified diuretic therapy is causal
7 Positive and worth mentioning is in any case the determination of urine electrolytes. In particular, the profound hypokalemia is likely to have made a decisive contribution to the clinical symptoms of the patients. In the case of metabolic alkalosis, this would be all the more desirable as urine chloride determination is now also mandatory. A low urine chloride is so far relevant and groundbreaking parameter
This point should therefore be clarified and emphasized once again, and I also recommend the following literature references: (Palmer and Clegg 2019)

Bohmig, G. A., S. Schmaldienst, W. H. Horl and G. Mayer (1999). "Iatrogenic hypercalcaemia, hypokalaemia and metabolic alkalosis in a lady with vena cava thrombosis--beware of overzealous diuretic treatment." Nephrol Dial Transplant 14(3): 782-784.

Khow, K. S., S. Y. Lau, J. Y. Li and T. Y. Yong (2014). "Diuretic-associated electrolyte disorders in the elderly: risk factors, impact, management and prevention." Curr Drug Saf 9(1): 2-15.

Palmer, B. F. and D. J. Clegg (2019). "The Use of Selected Urine Chemistries in the Diagnosis of Kidney Disorders." Clin J Am Soc Nephrol 14(2): 306-316.

Author Response

 A case of milk-alkali syndrome due to loop diuretic-induced alkaline load and polypharmacy by Naoya Mizutani and colleagues is a good and instructive review paper that exemplifies well the potential consequences of polypharmacy especially in the elderly. However, individual points should be reiterated:

Response:
We appreciate you taking the time to review our manuscript and your valuable feedback.

  1. The determination of the chloride concentration in serum and urine is indispensable for the differential diagnosis of metabolic alkalosis (and basically also for urine ammonium excretion).

    Response:
    Thank you for your comment. We have added the recorded value of serum urine chloride upon admission and on day 2 in Table 1. Hypochloremia in this case indicated chloride depletion and maintenance of contraction alkalosis. However, in the present study, we did not examine urine ammonium excretion: if we had, we would have been able to determine whether hypokalemia existed as another maintenance factor for metabolic alkalosis.

    We have added the following sentence regarding hypochloremia (lines 95–96):

    “The patient’s hypochloremia was indicative of chloride depletion and contraction alkalosis.”

  2. The amount and type of infusion therapy (ideally in milliliters per kilogram of body weight and the time period) should be added. A follow-up regarding the urine could drip would be desirable.

    Response:
    Thank you for your comment. We have added Figure 2 in response to your comments. The figure depicts the course of treatment in this patient with the dynamic values and is cited in line 104.
    We did not perform a follow-up examination of urine electrolytes in this case.
    After correction was completed, the serum electrolytes were stable within the reference range, so the urinalysis was not retested.

  3. line 107: instead of "transfusion" --> infusion should be meant.

    Response:
    Thank you for your comment. We have replaced “transfusion” with “infusions” in line 103.

  4. in the discussion it should be mentioned that especially elderly people with acute renal deterioration are considerably vulnerable to higher electrolyte shifts (Khow, Lau et al. 2014).

    Response:
    Thank you for your comment and for the reference suggestion. We have added the following information in the text, along with the supporting citation as reference number 2 (lines 140–142):

    “Aging causes changes to many organs, including the kidney, which maintains electrolyte balance. Diuretic use may impair the ability to maintain homeostasis, resulting in susceptibility to electrolyte and acid-base disorders [5].”

  5. line 137/138: I think this sentence should be deleted. Loop diuretic therapy has been shown to force contraction alkalosis, but it has not been adequately or sufficiently proven as the sole cause (even in this case). Moreover, there are other publications that shed more light on this point of view (Bohmig, Schmaldienst et al. 1999).

    Response:
    Thank you for your valuable comment. According to reference number 12, which you suggested, thiazides can cause volume depletion and hypokalemia, which are accompanied by hypercalcemia without excess calcium intake, hyperparathyroidism, or elevated vitamin D levels.
    We have added the following information in the text, along with the supporting citation as reference number 12 (lines 155–157):

    “Thiazides can also cause profound hypercalcemia without excess calcium intake, hyperparathyroidism, or elevated vitamin D levels [13].”

    In addition, we have deleted the following sentence as per your comment: However, to the best of our knowledge, this is the first report of loop diuretic-induced milk-alkali syndrome.
    Instead, we have added the following sentence in the Discussion section, considering the recommendation by Reviewer 1 to reduce the emphasis on furosemide as a cause of milk-alkali syndrome in this case (lines 138–139):
    However, to the best of our knowledge, this is the first report of milk-alkali syndrome caused by polypharmacy including both loop diuretics and thiazides.

  6. the title should replace the term "alkaline load" by solely alkalosis, since pathophysiologically primarily the contraction alkalosis by the intensified diuretic therapy is causal

Response:
Thank you for your comment. We have revised the title as follows:

Original
 A case of milk-alkali syndrome due to loop diuretic-induced alkaline load and polypharmacy

Revised
 A case of milk-alkali syndrome caused by diuretics-induced alkalosis and polypharmacy

  1. Positive and worth mentioning is in any case the determination of urine electrolytes. In particular, the profound hypokalemia is likely to have made a decisive contribution to the clinical symptoms of the patients. In the case of metabolic alkalosis, this would be all the more desirable as urine chloride determination is now also mandatory. A low urine chloride is so far relevant and groundbreaking parameter

    This point should therefore be clarified and emphasized once again, and I also recommend the following literature references: (Palmer and Clegg 2019)

    Response:
    Thank you for your valuable comment. We have added the recorded value of serum urine chloride on admission and on day 2 of hospitalization in Table 1. We have also corrected the value of urinary K/Cr ratio to 25.4 mEq/gCr (line 75), as we had not multiplied the original value by 102 in order to convert it to SI units.
    We have also added the following information, along with supporting reference number 4, to show how we interpreted the results of urine electrolytes analysis as follows:

    Original
    “We suspected that her symptoms were drug-induced.”

    Revised (lines 94–101):
    “Venous blood gas analysis results revealed remarkable metabolic alkalemia. The patient’s hypochloremia was indicative of chloride depletion and contraction alkalosis. Further, the patient’s urine potassium/creatinine ratio was over 13 mEq/g·Cr, blood pressure was not high, and urine Cl was over 15 mEq/L, which was suggestive of active diuretic use or hereditary tubular disease such as Bartter syndrome or Gitelman syndrome [4]. Based on the history of the present illness, we suspected that the patient’s symptoms were likely to be caused by diuretic-induced hypokalemia with hypomagnesaemia, which maintains Cl-resistant metabolic alkalemia.”

References

[5] Khow KS, Lau SY, Li JY, Yong TY. Diuretic-associated electrolyte disorders in the elderly: risk factors, impact, management and prevention. Curr Drug Saf. 2014 Mar;9(1):2-15. doi: 10.2174/1574886308666140109112730. PMID: 24410347.

[13] Böhmig GA, Schmaldienst S, Hörl WH, Mayer G. Iatrogenic hypercalcaemia, hypokalaemia and metabolic alkalosis in a lady with vena cava thrombosis--beware of overzealous diuretic treatment. Nephrol Dial Transplant. 1999 Mar;14(3):782-4. doi: 10.1093/ndt/14.3.782. PMID: 10193842.

[4] Palmer BF, Clegg DJ. The Use of Selected Urine Chemistries in the Diagnosis of Kidney Disorders. Clin J Am Soc Nephrol. 2019 Feb 7;14(2):306-316. doi: 10.2215/CJN.10330818. Epub 2019 Jan 9. Erratum in: Clin J Am Soc Nephrol. 2019 Aug 7;14(8):1241. PMID: 30626576; PMCID: PMC6390907.

Round 2

Reviewer 1 Report

The authors answered satisfactory to my comments.

Minor English editing needed.

Reviewer 2 Report

The authors have improved the manuscript once again, both scientifically and in terms of content. It is now a very good example of diuretic-induced alkalosis, which for internal medicine physicians can substantially improve their work.  I am satisfied and recommend the publication.